# Effect of Cutting Fluid on Milled Surface Quality and Tool Life of Aluminum Alloy

**DOI:** 10.3390/ma16062198

**Published:** 2023-03-09

**Authors:** Shuoshuo Pang, Wenxiang Zhao, Tianyang Qiu, Weiliang Liu, Pei Yan, Li Jiao, Xibin Wang

**Affiliations:** 1School of Mechanical Engineering, Beijing Institute of Technology, No. 5 Zhongguancun South Street, Haidian District, Beijing 100081, China; 2Key Laboratory of Fundamental Science for Advanced Machining Beijing Institute of Technology, No. 5 Zhongguancun South Street, Haidian District, Beijing 100081, China

**Keywords:** cutting fluid, aluminum alloy, tool life, cutting force, surface quality

## Abstract

The machining process of aluminum alloy usually produces built-up edge and tool sticking problems due to their low hardness and large plastic deformation, which may further affect the machined surface quality and tool life. This paper aims to investigate the influence of different cutting fluids on the machined surface quality and tool life during the milling process of 7050 aluminum alloy. A novel cutting fluid (QC-2803) was considered in the study, which is synthesized by addition of alkyl alcohol amide and chlorinated polyolefin, and the traditional cutting fluid (CCF-10) was used as the control group. The physical and chemical properties of two cutting fluids were characterized. The milling process of 7050 aluminum alloy was carried out under two different cutting fluid conditions. The machined surface morphology, cutting force and tool wear morphology were observed during the process. Results show that the surface tension of the novel cutting fluid is significantly lower than that of the traditional cutting fluid, which makes it easier to produce a lubricating film between the aluminum alloy and tool, and further benefits the machined surface quality and tool life. As a result, the surface roughness and cutting force are reduced by ~20.0% and ~42.9%, respectively, and the tool life is increased by 25.6% in the case of the novel cutting fluid (QC-2803). The results in this paper revealed the important laws of cutting fluid with metal surface quality, cutting performance and tool wear, which helps to control the machined surface quality and tool life by the selection of cutting fluid during metal milling.

## 1. Introduction

Aluminum alloys have been widely used in aerospace, aviation and other fields due to their high strength, low density, and corrosion resistance [1]. However, the melting point of aluminum alloy is low, and their plasticity increases with the increase of cutting temperature. In the condition of high temperature and pressure, the friction at the cutting interface becomes great, so it is not easy to obtain high surface quality. High-speed machining technology offers many advantages over conventional machining, such as higher material removal rate, increased machining accuracy and better surface finish [2], and these advantages have attracted numerous interests because of its strong vitality and broad application prospects [3]. Generally, the material removal rate reaches 85 cm^3^/min during the machining process of thin-walled aluminum alloy components and structures for large aircraft, and high tool wear resistance is required. The milling process of aluminum alloy is an intermittent machining process, and the tool is subjected to cyclic impact when cutting in and out, which will aggravate tool wear. Tool wear will further affect the machining quality of the workpiece. Therefore, it is essential to study the tool wear during the machining process of aluminum alloy to improve machined surface quality and tool life.

The aeronautical aluminum alloy machining process usually produces built-up edges and tool sticking problems due to their low hardness and large plastic deformation. In addition, fusion welding usually occurs on the cutting edge, which may further affect cutting efficiency, surface roughness and machining accuracy of aluminum alloys. Cutting fluids have good cooling and lubricating properties; it is commonly used in metal cutting to effectively reduce machining temperatures, promote chip evacuation, improve cutting efficiency and surface quality [4]. According to the statistics, costs related to cutting fluid account for approximately 17% of the cost of the finished workpiece, while the cost of the tool only accounts for 4%. At present, water-based cutting fluids are gradually replacing oil-based cutting fluids due to the shortage of oil and high cost [5,6,7].

Currently, research on cutting fluid is mainly divided into three aspects: the application method of cutting fluid, its influence on machining, and the development of environment-friendly cutting fluid. During cutting process, to reduce cutting temperature and prolong tool life, large amounts of cutting fluid are used, which may damage for the environmental pollution and human health [8,9]. New application methods of cutting fluid have been widely studied to address the above problems. The common application methods include Minimum Quality Lubrication (MQL) [10,11,12,13], High-Pressure Coolant (HPC) and Little Quantity Lubricant (LQL). Zenjanab et al. [14] investigated the effect of CuO-nanofluid with 0.25, 0.5 and 1% boric acid nanoparticles on cutting forces, surface roughness and tool wear during high speed turning of hardened AISI 4340 steel, and the results showed that 1 wt.% volume fraction of boric acid-nanoparticles added to soluble oil as cutting fluid significantly reduced surface roughness and tool wear, and reduced cutting forces by one quarter compared to dry cutting. Diniz et al. [15] studied the effect of cutting fluid pressure on tool wear at different cutting speeds in machining processes, and results showed that the use of a lower cutting speed (Vc = 490 m/min) in dry cutting leads to a tool life close to that obtained with cutting fluids, but when the cutting speed was increased (Vc = 570 m/min), high-pressure coolant can effectively prolong the tool life. Avila et al. [16] performed continuous cutting experiments of AISI 4030 under different cutting fluids and measured the efficiency through cutting tool life and workpiece surface finish; results showed that reducing in the cutting fluid concentration from 5 to 3% resulted in lower tool life, particularly at a cutting speed of 300 m/min.

Jayal et al. [17] studied the effect of different cutting fluid types and application modes on tool wear at different cutting speeds, feeds and hole depths; results showed that dry cutting had the lowest hole accuracy, spray lubrication had the worst aerosol concentration and synthetic cutting fluids gave the highest dimensional accuracy and aerosol concentration. Su et al. [18] investigated the tool wear under different cooling and lubricating condition with a coated carbide tool at a cutting speed of 400 m/min; the experimental results showed that diffusion wear and thermal fatigue wear are the main wear mechanisms of coated tools under various cooling/lubrication conditions, compressed cold nitrogen gas and oil mist (CCNGOM) can effectively improve tool life. Yang et al. [19] studied the effect of a new water-based cutting fluid on tool wear and cutting forces in milling titanium alloy, and experimental results showed that the newly developed cutting fluid reduces cutting forces and tool wear, and achieves a smaller surface roughness (Sa = 0.563 µm) compared to commercial emulsions. Debnath et al. [20] compared the effects of different cutting fluid levels and cutting parameters on surface roughness and tool wear; the article applied extreme difference analysis, and the experimental results showed that feed rate had the greatest effect on surface roughness (34.3%), cutting speed had the greatest effect on tool wear (43.1%), and cutting fluid had a significant effect on surface roughness (33.1%) and tool wear (13.7%), the optimum parameters of surface roughness and tool wear were obtained at high cutting speed (180 m/min), medium depth of cut (1 mm), low feed rate (0.05 mm/rev) and under the conditions of low flow rate high velocity (LFHV) cutting fluid selection. Khandekar et al. [21] studied the tool wear, cutting force, roughness of workpiece surface and chip thickness of dry cutting, conventional cutting fluid and nano-cutting fluid (adding 1% Al_2_O_3_ nanoparticles), and the results showed that compared with dry cutting and conventional cutting fluid, the cutting force was reduced by ~50.0% and ~30.0%, and the Ra value of the nano-cutting fluid processing surface was reduced by ~54.5% and ~28.5%, respectively, which prolonged the tool life. Su et al. [22] studied the influence of different cutting fluid pressure (3~6 MPa) on tool wear under different cutting speeds (100~300 m/min) and dry cutting through processing compacted graphite iron (CGI), and the results showed that high-pressure cutting fluid can effectively reduce tool wear, and the tool life under 6 MPa high pressure is almost twice that of dry cutting. Khanna et al. [23] compared the processing performance (cutting temperature, cutting force, tool wear and surface roughness) for dry and liquid carbon dioxide (LCO_2_) cutting methods, and the results indicated that compared with dry machining, the cutting zone temperature under LCO_2_ cutting conditions is reduced by 80%, tool wear is reduced by 5%, and cutting force modulus is reduced by 5%. Pereira et al. [24,25] proposed a cooling lubrication system by combining CO_2_ cryogenic and MQL technique, and the results showed that tool life was extended to 50% in MQL + CO_2_ machining process compared with dry cutting, MQL and CO_2_ cryogenic individually.

In addition, a large number of experimental studies have been carried out on vegetable oil as a metalworking cutting fluid, and it is found that vegetable oils can reduce friction and improve tool life in metal processing compared to mineral oils [26,27,28,29]. Pereira et al. [30] studied environmentally friendly cutting fluids, compared four vegetable oils and one ECO-350 recycled oil, and through the analysis of tribological and rheological properties, the results showed that ECO-350 recycled oil improved tool life by 30% compared to commercial vegetable oil. Zhang et al. [31] analyzed the effects of different vegetable oils on the cooling performance of MQL abrasive grains experimentally, and the physical properties (viscosity and surface tension) of vegetable oils; results showed palm oil nanofluid with a high viscosity and surface tension achieved the lowest friction coefficient (0.258), specific friction energy (27.09 J/mm^3^), and grinding peak temperature (123.8 °C) and exhibited better grinding performance than the others.

Moreover, current research shows cutting fluids with extreme pressure additives can effectively improve the lubricating adhesion of the cutting tool under high temperature, workpiece surface finish and tool life [32]. Cambilella et al. [33] studied the effect of the concentration of different emulsifiers (anionic, non-ionic and cationic surfactants) on the extreme pressure properties of oil–water emulsion, and the results showed that the lubricating performance depends to a great extent upon the type and concentration of emulsifier. Jain et al. [34] carried out the friction and wear experiments for various inorganic metal chlorides, sulfates and phosphates on four-ball wear and EP testing machine, and results showed that the interaction of zinc sulfate–zinc chloride and aluminum sulfate–zinc chloride is beneficial to improving the adhesion property. Wang et al. [35] proposed a new method for mixing vegetable oil additives, through the addition of sulfur-based extreme pressure (EP) additives to canola oil to improve the lubrication and cooling performance of the machining area, the results showed that the combination of canola oil + sulfur-based EP additives, which effectively decreased the temperature of the cutting area and wear of cutting tools, the inorganic film produced by the EP additive molecule helps prevent direct contact between the tool and the workpiece, reducing tool wear and improving surface quality. Maruda et al. [36] studied the influence of extreme pressure and anti-wear (EP/AW) additives on the surface topography of double-phase steel during turning with different cooling media and variable flow rates, and results showed that, in comparison with dry cutting, phosphate-based EP additive takes shape in the high-concentration anti-wear friction film on the machined surface, decreasing friction in the contact zone between the cutting tool and the workpiece, thereby lowering the parameters of selected surface topography by 6–38% and the wear of the flank surface by 23%. In addition, Maruda et al. [37] studied tool wear of P25 cemented carbide inserts in finish turning of AISI 1045 carbon steel for different cooling conditions: dry cutting, minimum quantity cooling–lubrication (MQCL) and MQCL with phosphate ester-based EP/AW additive, and results showed that the wear of the inserts using the MQCL + EP/AW method is reduced by about 40% compared to dry cutting and about 25% compared with MQCL.

The purpose of this paper is to investigate the influence of the novel water-based cutting fluid and the traditional cutting fluid on the machined surface quality and tool life during the milling process of 7050 aluminum alloy. Firstly, the physical and chemical properties of two cutting fluids were characterized, which is synthesized by addition of alkyl alcohol amide and chlorinated polyolefin. Then, the effects of cutting fluids on the machined surface quality and tool life were investigated, including cutting force, surface roughness, morphology and tool wear.

## 2. Materials and Methods

### 2.1. Workpiece and Tool Materials

The workpiece material used in the study is high-quality 7050 aluminum alloy that has been heat-treated (Solution-ageing, HB = 135), and the size of the workpiece is 80 mm × 76.5 mm × 40 mm. The chemical composition of 7050 aluminum alloy is shown in Table 1. The tool used in the study was a three-edge straight shank uncoated carbide end milling cutter (XS2111-075K-GNS, SG, Shanghai, China) with a diameter of 10 mm and a corner radius of 0.25 mm.

### 2.2. Cutting Fluids

In this study, milling experiments were carried out under two different cutting fluid conditions. The traditional one is micro-emulsion semi-synthetic cutting fluid (CCF-10, Tellent, Tianjin, China), which is a kind of water-soluble emulsified cutting fluid containing lubricating oil composition. The novel one is a type of water-based cutting fluid (QC-2803), which is developed by the Tianjin Equipment Research Institute of Tsinghua University. The components and functions of these two cutting fluids are shown in Table 2. It should be noted that 5% of the original cutting fluid were mixed with 95% of deionized water to prepare the cutting fluid required for the milling experiment.

In order to investigate the cooling and lubricating properties, physical and chemical properties of these two cutting fluids were characterized, including viscosity, surface tension, cooling curve, pH and friction coefficient. The viscosity of cutting fluid was measured by rotary rheometer (TA HR-2, DE, USA), surface tension was calculated by surface tensiometer (Dataphysics DCAT 21, Stuttgart, Germany), and the pH was tested by conductivity meter (InoLab Cond 740, Munich, Germany). Friction and wear tests were carried out to obtain the friction coefficients of two cutting fluids, which were measured by Universal Micro-Tribotester (UMT-2, CA, USA). The cooling performance of the cutting fluid is based on the method of measuring the heat transfer coefficient of quenching oil.

### 2.3. Milling Experiment

The purpose of this study is to compare the effects of two cutting fluids on the machined surface quality and tool life during the milling process of 7050 aluminum alloy. The three-axis machining center with Spindle speed of 20,000 r/min, *x*, *y*, *z*-axis travel of 600 × 400 × 300 mm, positioning accuracy of ±0.005 mm and motor power of 5.5 KW (TV600, CNC, Shandong, China) was used, and the adopted cutting parameters are illustrated in Table 3 (data provided by industrial company). A novel external circulation system was designed for using the cutting fluids conveniently. Figure 1 shows the schematic diagram of the entire machining system. The milling test was repeated three times and then the average of the tool wear was used as the result.

The workpiece was, firstly, milled to obtain a plane of 80 mm × 76.5 mm, and then milling experiments were performed with the adopted cutting parameters under two cutting fluid conditions. The total milling time for each cutting fluid was 200 min, and cutting force, surface quality as well as tool wear were measured and recorded at the designed time intervals. Kistler9257 dynamometer (Range: ±5 KN, table size: 100 × 170 mm, response frequency: 1 kHz) was used to collect cutting force (cutting forces were measured 42 times at 5-min intervals, including the first 4 times in the first 10 min), and Keyence VK-100 laser 3D scanning microscope was used to measure the surface roughness and surface topography of the workpiece. CCD camera and JSM-7200F Thermal Field Emission Scanning Electron Microscopy (SEM) with Through-The-Lens System (TTLS) were used to record the tool wear morphology.

## 3. Results

### 3.1. Physical and Chemical Properties of Cutting Fluid

#### 3.1.1. Viscosity

Viscosity is a basic physical property that describes a fluid, which plays an important role in the lubricating and film formed penetrating ability of the cutting fluid, and it affects the thickness of the lubricating film formed between the tool and workpiece surface. The dynamic viscosity of two cutting fluids is shown in Figure 2. It can be seen that the viscosity is 0.00134 Pa·s and 0.00109 Pa·s for QC-2803 and CCF-10, respectively. The viscosity of QC-2803 is 22.9% higher than that of CCF-10, and it can be assumed that in the actual aluminum alloy milling process, the lubricating film formed by QC-2803 has higher strength and better friction reduction performance than CCF-10.

#### 3.1.2. Surface Tension and pH

The surface tension of the liquid affects the permeability of the liquid, and it can be used to characterize the ability of the cutting fluid to spread out and form a lubricating film. The pH value reflects the corrosive properties of the cutting fluid. As shown in Figure 3, the surface tension of QC-2803 and CCF-10 is 30.7 mN/m and 31.8 mN/m, respectively. Compared with water, the surface tension of the two cutting fluids is smaller, and it can be seen that the addition of surfactants to the cutting fluid significantly reduces the surface tension. The surfactant mainly plays the role of emulsification in the cutting fluid. One end of the surfactant molecule is hydrophilic and the other end is lipophilic, which can combine water and oil and significantly reduce the surface tension of the cutting fluid. The alkyl alcohol amide in QC-2803 has a better effect on reducing the surface tension than sodium petroleum sulfonate in CCF-10. Table 4 shows that the pH value of QC-2803 and CCF-10 is 9.7 and 9, respectively. The pH value of the two cutting fluids is not much different, and both are within the range specified in the standard, and will not cause irritation and allergy to workers.

#### 3.1.3. Cooling Performance

The cooling performance of the cutting fluid can effectively take away the cutting heat generated, which contributes to reducing the cutting temperature and improving the machined surface quality. The cooling curves of two cutting fluids are shown in Figure 4. The result shows that the cooling performance of QC-2803 is slightly better than that of CCF-10, and the same conclusion can be drawn from the above analysis of surface tension.

#### 3.1.4. Friction Coefficient

The friction coefficient results of QC-2803 and CCF-10 are shown in Figure 5. The friction coefficient of CCF-10 is around 0.12, and the friction coefficient of QC-2803 is around 0.1. The friction coefficient of QC-2803 cutting fluid is larger than that of CCF-10 (increased by 20.0%). It is because QC-2803 contains extreme pressure additive chlorinated polyolefin, which can maintain a significant anti-friction effect under high contact stress and effectively reduce the friction force [28]. Therefore, it can be concluded that QC-2803 promotes the formation of a lubricating film and reduces friction between the workpiece and the tool.

### 3.2. The Effect of Cutting Fluid on Surface Quality

#### 3.2.1. Surface Roughness

The variation curve of surface roughness against cutting time is shown in Figure 6. It can be seen that both surface roughness curves show a gradually decreasing trend with the increase of cutting time. Overall, the surface roughness value of the novel cutting fluid is significantly lower than that of the traditional cutting fluid, and the surface roughness of CCF-10 produce a more severe fluctuation than QC-2803 cutting fluid. The maximum surface roughness is around 3.5 μm and 2.6 μm for CCF-10 and QC-2803, and the minimum surface roughness is around 2.0 and 1.6 for CCF-10 and QC-2803, respectively. As a result, the surface roughness in the case of QC-2803 condition is reduced by 20.0% on average, compared with CCF-10.

#### 3.2.2. Surface Morphology

The morphology results of the machined workpiece surface are shown in Figure 7 and Figure 8. Figure 7 shows the 3D surface morphology at the beginning and end of machining time under two cutting fluid conditions. In combination with Figure 6, as the wear of the tool increases, surface roughness reduces with machining time. The back angle of the tool becomes smaller as the tool wear increases. Then, the extrusion of the tool and the machining surface becomes stronger, and the temperature generated during processing increases, which promotes the softening of the material, thereby the surface roughness is reduced. Figure 8 shows that there are similar concentric lay patterns on both machined surfaces, but the distance between adjacent milling tool marks seem longer in the case of QC-2803 cutting fluid. In addition, the surface roughness of CCF-10 cutting fluid is 1.96 μm, while the surface roughness of QC-2803 cutting fluid is 1.6 μm. The results showed that the new cutting fluid can significantly reduce the roughness of the processing surface and improve the quality of the processing surface.

### 3.3. The Effect of Cutting Fluid on Tool LIFE

#### 3.3.1. Cutting Force

The original signals of cutting force were collected in three directions of X, Y, and Z, as shown in Figure 9. The results of the three cutting force components of post-processing are shown in Figure 10. The maximum cutting force occurs at the beginning when cutting into the workpiece (including impact [38,39,40]). Overall, the cutting force amplitude in Z direction is significantly smaller than those in X and Y directions, and the cutting force fluctuation is small. Both cutting force amplitude and fluctuation in case of QC-2803 are dramatically smaller than those in case of CCF-10. In X direction, the maximum cutting force amplitude is around 120 N and 80 N in case of CCF-10 and QC-2803, respectively. In Y direction, the maximum cutting force amplitude is around 90 N and 40 N in case of CCF-10 and QC-2803, respectively. In the direction of the real cutting force (resultant force F), the maximum cutting force amplitude in case of CCF-10 and QC-2803 is around 150 N and 90 N, respectively. It suggests that QC-2803 cutting fluid can reduce amplitude and fluctuation of cutting force. Cutting forces affects the surface finish and dimensional accuracy of the machined structure. With the increase of machining time, tool wear increases, the fluctuation amplitude of cutting force becomes larger, and the roughness of the machining surface fluctuates greatly, as shown in the analysis of Section 3.2.1. The surface roughness remains stable during the processing process of QC-2803 cutting fluid, indicating that the tool milling ability is more stable compared with CCF-10 cutting fluid.

#### 3.3.2. Tool Life

In order to study the effect of the novel water-based cutting fluid on tool life in the milling process of aluminum alloy, cutting fluid type was set as an independent variable while other milling parameters remain unchanged. The experimental results showed that the tool wear mode is nose-damage when the milling starts. QC-2803 cutting fluid can greatly reduce the degree of nose damage and prolong tool life. In the experiment, the GAOSUO 1003+ digital camera was used to examine the tool wear. Table 5 shows the tool wear amount of three edges at designed time intervals (0 min, 3 min, 8 min, 60 min, 120 min and 200 min) under CCF-10 and QC-2803 cutting fluid conditions. Figure 11 shows tool wear results under two cutting fluid conditions. It can be seen that the wear amount of three edges increases against cutting time at different rates under different cutting fluid conditions. Compared with the CCF-10 cutting fluid, the wear rate and amount in case of QC-2803 is slower and smaller. As shown in Figure 11a,b, the tool wear is developing to the third stage in the condition of CCF-10 cutting fluid, while it is still in the gentle wear stage in condition of QC-2803 cutting. During milling time of 200 min, the maximum cutting-edge wear of CCF-10 cutting fluid and QC-2803 cutting fluid is 0.262 mm and 0.195 mm, respectively. It can be calculated that the maximum cutting-edge wear of QC-2803 cutting fluid is 74.4% of that in the case of CCF-10 cutting fluid.

Figure 12 shows the SEM image of the entire tool wear surface after machining under two cutting fluid conditions. Figure 13 and Figure 14 show SEM images of cutting-edge wear for CCF-10 and QC-2803, respectively. It can be seen that the tool failure mode behaves damage and crater wear at the nose of tool. Under the two cutting fluid conditions, the tool possesses different degrees of damage and wear. Figure 15 shows EDS analysis of tool wear after machining, (a) and (b) represent EDS analysis at tool wear under CCF-10 and QC-2803 cutting fluid conditions respectively. The results showed the presence of tool base elements (C, W, O) and Al, which indicates that there is adhesive wear [18,19,21]. In the case of QC-2803 cutting fluid, the three-edge damage degree, damage volume and crater wear area of the tool are obviously smaller than those of CCF-10.

## 4. Discussion

The viscosity and friction coefficient results of two cutting fluids show that the lubricating performance of QC-2803 cutting fluid is superior to that of CCF-10, and the surface tension and cooling curve results illustrate that the cooling effect of QC-2803 cutting fluid is better. The big differences between two cutting fluids are lubrication, and the ability to form a lubricating film directly influence the machined surface quality and tool life.

During the milling process, the tool and the workpiece keep in continuous contact, friction and frictional heat generated in the relative motion play important roles in the surface quality of workpiece [41]. Moreover, the aluminum alloy is soft material, which makes it easy to stick to the tool during milling, and then results in an increase in the surface roughness [42]. From the above surface roughness and morphology analysis, it can be seen that QC-2803 cutting fluid produced a smoother machined surface compared to CCF-10 cutting fluid, because the QC-2803 cutting fluid has better anti-friction and lubricating performance than CCF-10. Therefore, it is easier to form a lubricating film with a certain strength between the tool and the workpiece surface, reduce the friction between tool and the machined surface effectively, further reduce surface roughness and improve the machined surface quality [43]. The cooling performance of QC-2803 cutting fluid is better than that of CCF-10 cutting fluid, and it benefits to reduce the temperature during milling and improve the machined surface quality.

At the beginning of milling process, the cutting motion works from the fillet, so the cutting force is the largest (including impact) due to vibration when cutting into the workpiece, as shown in Figure 8 and Figure 9. Previous literature results point out that the milling cutter is subject to impacts when cutting motion starts and ends. When cutting motion ends, the cutting edge at the fillet cuts out firstly and bears the main impact force. The cutting edge is repeatedly subjected to the continuous cut-in and cut-out impact force [44]. Therefore, the tool damage is most likely to occur on cutting edge at the arc, including edge micro-damage and damage. As shown in Figure 12 and Figure 13, the tool wear is mainly damage, and the degree of damage in the case of CCF-10 cutting fluid is greater than that of QC-2803. It is because the cutting fluid QC-2803 with good lubricating and cooling properties more easily forms a lubricating film between the tool and the workpiece and further mitigates tool wear, which agrees with the results in the literature [45]. The lubricating film is good for reducing the friction and frictional heat generation, and weakening the surface work hardening. All these results suggest that QC-2803 cutting fluid has great potential in reducing the amplitude and fluctuation of cutting force, tool damage, prolonging tool life and improving the machined surface quality.

## 5. Conclusions

In this paper, two cutting fluids were applied in the milling process of 7050 aluminum alloy. The cooling and lubricating properties of cutting fluids were characterized. The influence of cutting fluids on the machined surface quality and tool life were investigated. The main achievements are as follows:In terms of the physical and chemical properties of the cutting fluid, the viscosity of QC-2803 is 22.9% higher than that of CCF-10, the surface tension of QC-2803 is about 96.5% of CCF-10, the friction coefficient of QC-2803 increased by 20.0% compared with CCF-10. It is easier to form a lubricating film with a certain strength in case of QC-2803, and further reduce the friction between the tool and the workpiece.QC-2803 cutting fluid can form a stronger oil film and has better cooling performance compared with CCF-10 due to the extreme pressure additive components in water-based cutting fluid. The cutting force and its fluctuation are significantly decreased in both X and Y directions in case of QC-2803 cutting fluid. The cutting force is approximately reduced to 42.9% of CCF-10 cutting fluid.Higher surface quality is obtained in the milling process of aluminum alloy under the QC-2803 cutting fluid condition. The surface roughness is reduced by 20% on average. The maximum edge wear is 74.4% of that of CCF-10 cutting fluid, and it is assumed that the tool life under QC-2803 cutting fluid conditions is longer.

## Figures and Tables

**Figure 1 materials-16-02198-f001:**
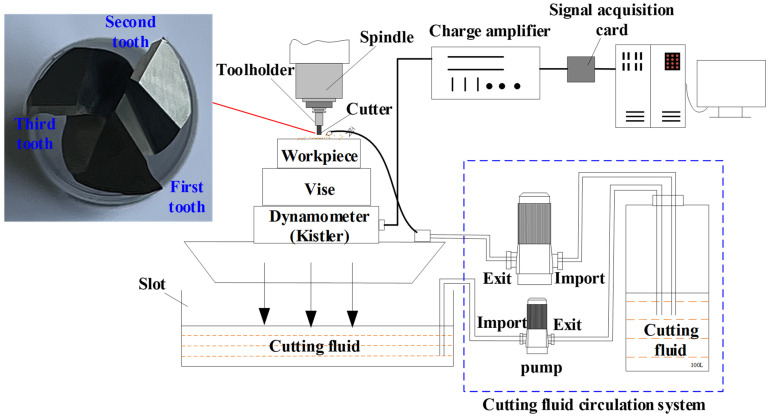
Schematic diagram of the entire machining system.

**Figure 2 materials-16-02198-f002:**
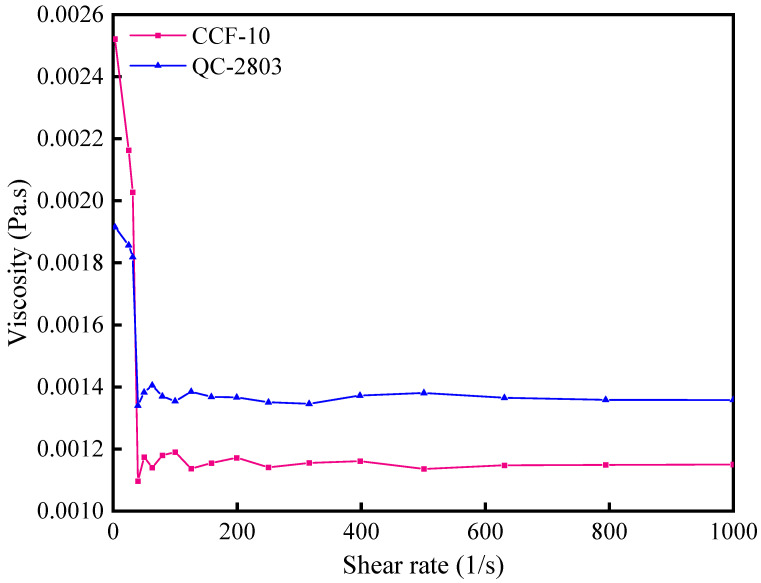
Viscosity of QC-2803 and CCF-10.

**Figure 3 materials-16-02198-f003:**
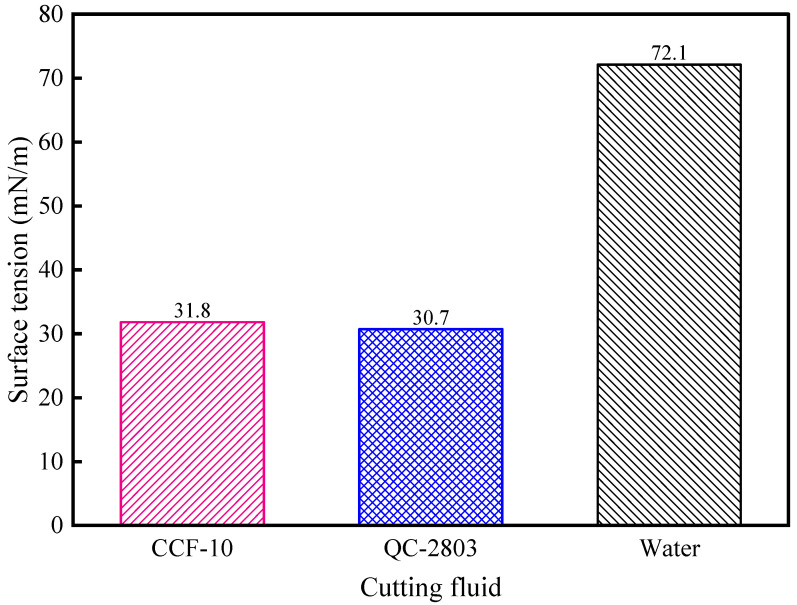
Surface tension for two cutting fluids.

**Figure 4 materials-16-02198-f004:**
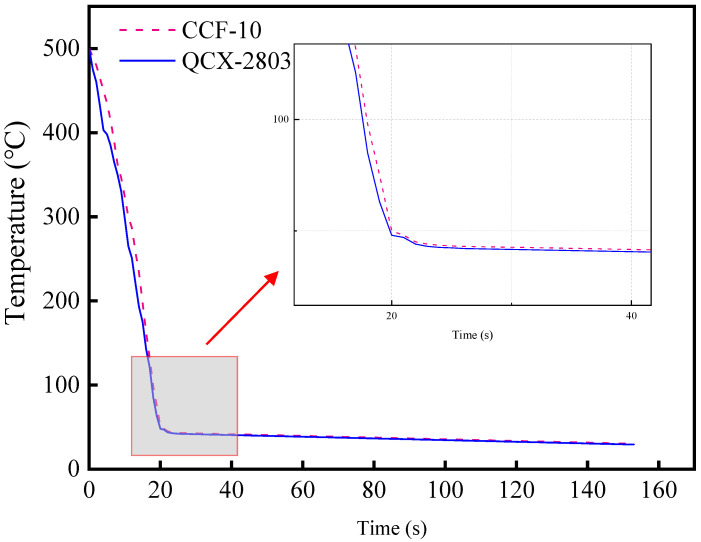
Cooling curve of CCF-10 and QC-2803.

**Figure 5 materials-16-02198-f005:**
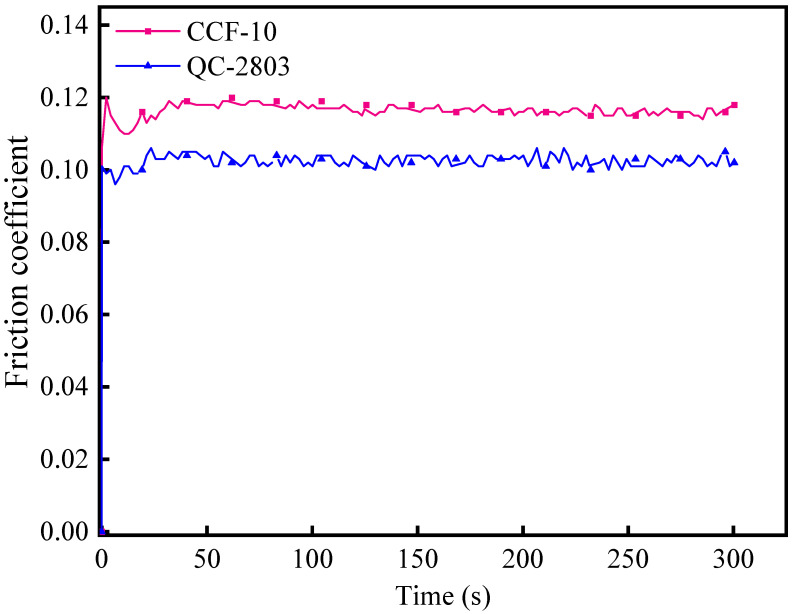
Friction coefficients of CCF-10 and QC-2803.

**Figure 6 materials-16-02198-f006:**
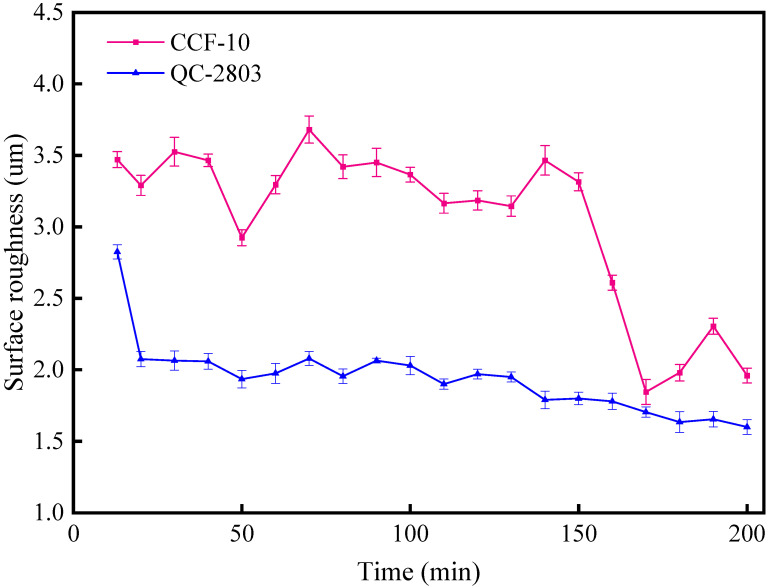
Surface roughness of CCF-10 and QC-2803.

**Figure 7 materials-16-02198-f007:**
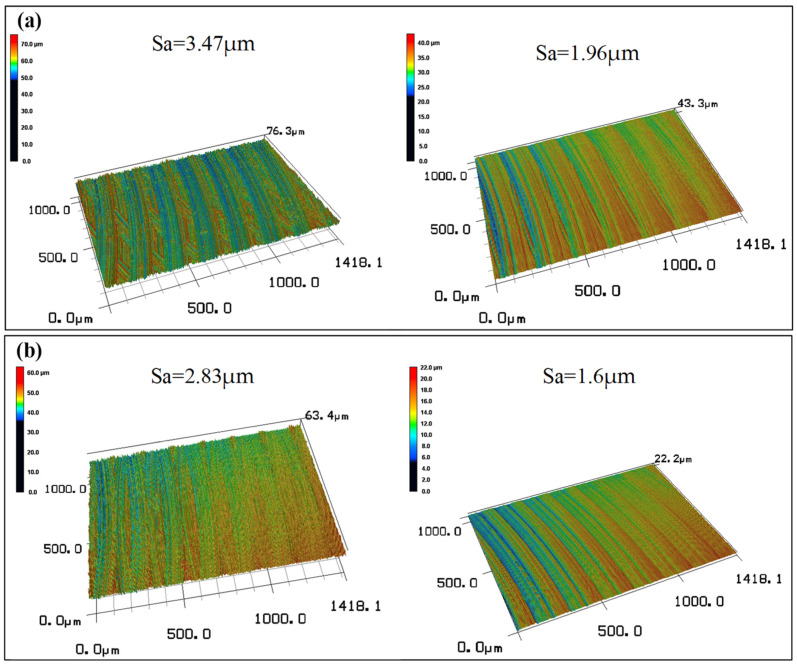
3D surface morphology at the beginning and end of machining time under two cutting fluid conditions: (**a**) CCF-10 (**b**) QC-2803.

**Figure 8 materials-16-02198-f008:**
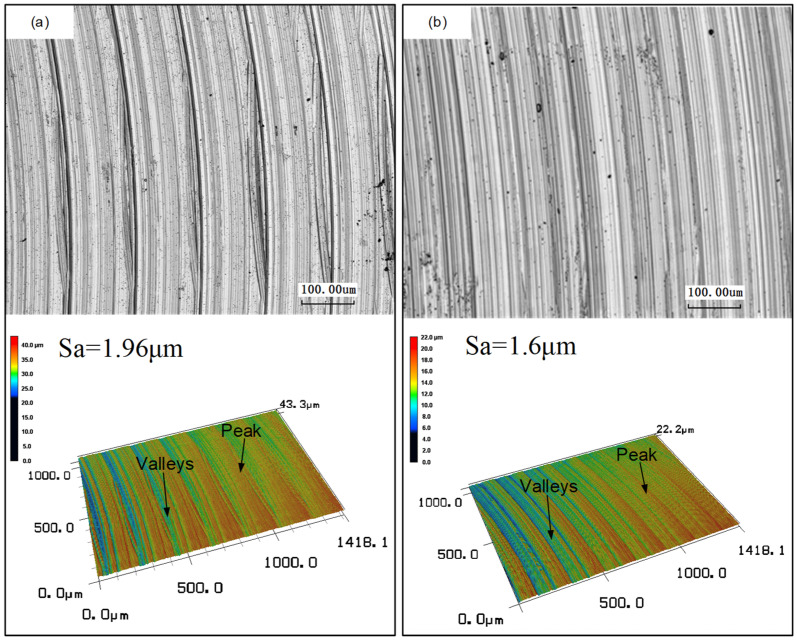
2D and 3D surface morphology under two cutting fluid conditions: (**a**) CCF-10 (**b**) QC-2803.

**Figure 9 materials-16-02198-f009:**
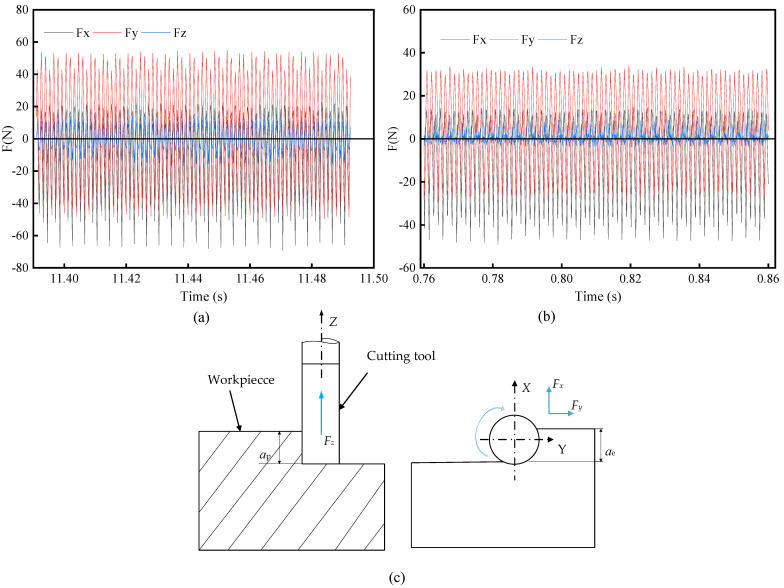
The original axial force signals during milling using (**a**) CCF-10 and (**b**) QC-2803; (**c**) cutting force direction.

**Figure 10 materials-16-02198-f010:**
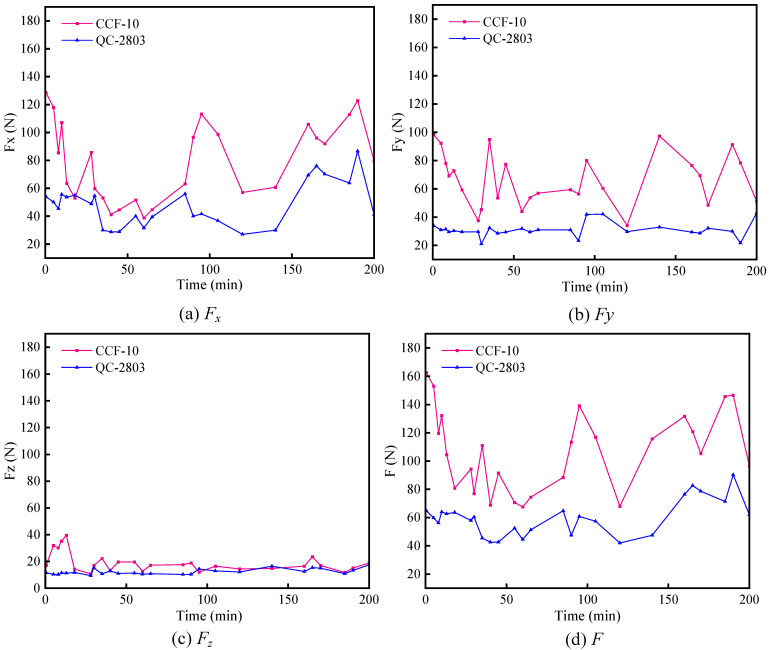
Cutting forces and its direction under two cutting fluid conditions.

**Figure 11 materials-16-02198-f011:**
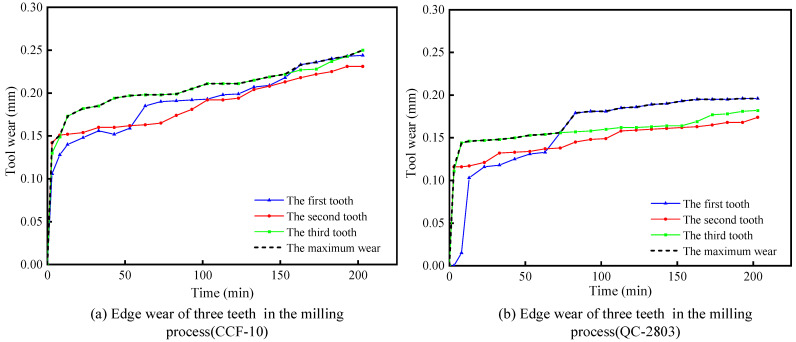
Edge wear in the milling process under two cutting fluid conditions.

**Figure 12 materials-16-02198-f012:**
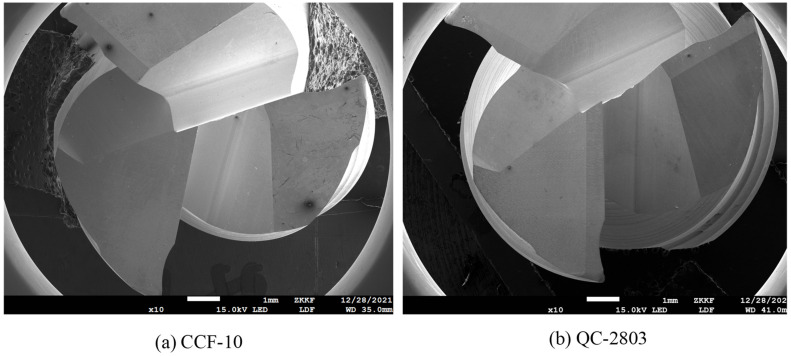
SEM images of wear surface of whole tool under two cutting fluid conditions.

**Figure 13 materials-16-02198-f013:**
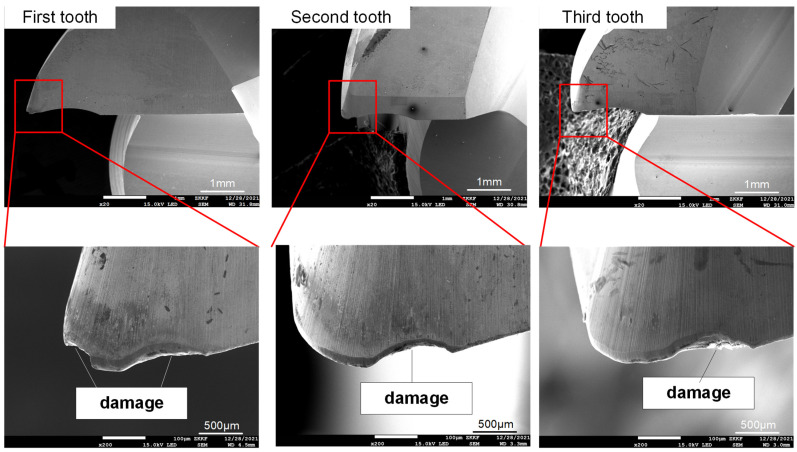
SEM images of cutting-edge wear in the case of CCF-10.

**Figure 14 materials-16-02198-f014:**
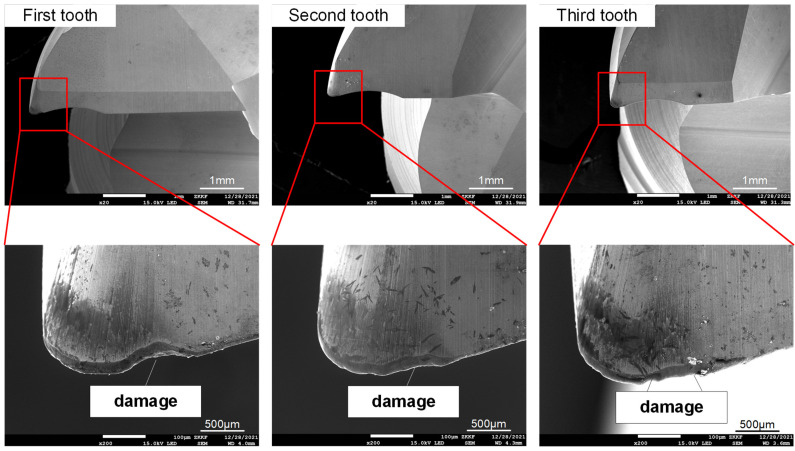
SEM images of cutting-edge wear in the case of QC-2803.

**Figure 15 materials-16-02198-f015:**
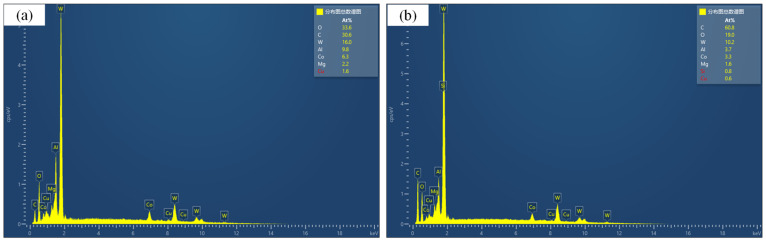
EDS analysisof cutting fluid (**a**) CCF-10, (**b**) QC-2803.

**Table 1 materials-16-02198-t001:** Chemical composition of 7050 aluminum alloy.

Element	Cr	Zr	Zn	Si	Fe	Mn	Mg	Ti	Cu	Al
Content (wt.%)	0.035	0.12	5.8	0.1	0.12	0.1	2.39	0.043	2.36	Rest

**Table 2 materials-16-02198-t002:** The main components and functions of two cutting fluids.

Components	CCF-10	QC-2803
Lubricant	Naphthenic oil	Tall oil
Diluent	Water	Water
Rust inhibitors	Methylbenzotriazole	Sodium Phosphate
Corrosion inhibitor	Sodium borate, zinc naphthenate	Benzotriazole, sodium borate
Surfactant	Fatty alcohol polyoxyethylene ether, Sodium petroleum sulfonate	Fatty Alcohol Polyoxyethylene Ether Sodium Hydroxylate, Alkyl Alcohol Amide
Antioxidant	2,2′-Methylenebis (6-tert-butyl-4-methylphenol)	2,4,6-Tri-tert-butylphenol
Extreme pressure lubricant	—	Chlorinated Polyolefin
Bactericides	—	Isothiazolinone
pH regularot	Diethanolamine	Triethanolamine

**Table 3 materials-16-02198-t003:** The cutting parameters of 7050 aluminum alloy.

Cutting Parameters	Cutting Speed (V, m/min)	Depth of Cut (a_p_, mm)	Cutting Width (a_e_, mm)	Feed Rate per Tooth (f_z_, mm/rev)
Value	471.000	0.300	9.000	0.067

**Table 4 materials-16-02198-t004:** pH values for two cutting fluids.

Cutting Fluid	CCF-10	QC-2803
pH	9.7	9.0

**Table 5 materials-16-02198-t005:** Changes in tool wear during milling (mm).

Cutting Fluid	CCF-10			QC-2803		
Time (min)	First Tooth	Second Tooth	Third Tooth	First Tooth	Second Tooth	Third Tooth
0	0	0	0	0	0	0
3	0.108	0.143	0.134	0	0.116	0.111
8	0.127	0.151	0.149	0.018	0.116	0.143
60	0.185	0.163	0.198	0.133	0.138	0.154
120	0.199	0.194	0.211	0.187	0.159	0.162
200	0.243	0.231	0.262	0.195	0.173	0.183

## Data Availability

The data that support the findings of this study are available from the corresponding author upon reasonable request.

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
