# Peer review of "Effect of Cutting Fluid on Milled Surface Quality and Tool Life of Aluminum Alloy"

_materials, 2023, doi:10.3390/ma16062198_

Round 1

Reviewer 1 Report

MANUSCRIPT ID - materials-2236398

The authors have presented a comprehensive comparative analysis of a cutting fluid which they claimed to have developed on their own with a commercial cutting fluid. The properties of the two fluids which have significant influence on their machining characteristics have been compared. Their machining performance is studied in terms of forces, surface roughness, and tool life. The study is interesting to readers in the area of metal cutting and tribology. However, the following corrections should be carried out.

1. “The material removal rate reaches more than 90% ---” - In the above statement in the Introduction section, check the units.

2. “Cutting fluid under extreme pressure can effectively improve the lubricating adhesion of the cutting tool under high temperature” - Please check this statement.

3. “Diniz et al. [24] studied the effect of cutting fluid pressure on tool wear at different cutting speeds in dry cutting” - Please check - dry cutting or wet cutting?

4. “The tool used in the study is three-edge straight shank carbide end milling cutters (XS2111-075K-GNS, SG, China) with a diameter of 10 mm and a radius of 0.25mm.” - Please check the radius value.

5. “The viscosity of cutting fluid was measured by rotary dynamometer (TA HR-2, US),”.  Is it a dynamometer or rheometer?

6. “A novel external circulation system was designed for the convenient exchange of cutting fluids.” Please change the word “exchange”.

7. “Viscosity is a physical parameter that measures the viscosity of a fluid….” Please correct.

8. “...thickness of the lubricating film formed between the cutting fluid and workpiece surface.”

9. Figure 4 and figure 5 are same. Figure 5 has to be friction coefficient values for the two cutting fluids.

10. What is the meaning of “The maximum surface roughness is around 3.5 and 2.6”? Why are two values provided. Also, mention the units.

11. What is the reason for reduction in surface roughness with machining time? The authors can compare the surface morphology at the beginning and end of machining machining time for the two fluids.

12. Milled surfaces will have concentric lay pattern as shown in figure 7 (b). How is that the lay pattern in figure 7 (a) has opposing (crisscrossing) concentric patterns near the areas marked as surface defects?

13. “It can be seen that the force in the X, Y, and Z directions has a drift phenomenon”. What do the authors mean by this statement?

14. What do the authors mean by “real cutting force”? How is it calculated?

15. Figure 10 (c) is not necessary. The maximum wear curves in (a) and (b) convey this information.

Reviewer 2 Report

Review of the manuscript 2236398

Title: Effect of Cutting Fluid on Milled Surface Quality and Tool Life 2 of Aluminum Alloy

Suggestion:

In the abstract:

use the same number of significant algharisms for the different data.

Consider the substitution of the word achievements for results.

Introduction:

First paragraph, try to correlate better the different information given, there is no textual cohesion, the authors jump from high speed machining, to aluminum alloys to milling…

The machining rates reach 90%, is there any literature to corroborate with those values? It is not usual.

Second paragraph, there are a lot of different references that can be used to corroborate with the first part of the paragraph. Please add those.

Line 46 there is a missing period symbol.

The statement made about cutting fluids are correct, however, it is more of the definition of the main cutting fluid application (to lubricate and/or refrigerate) and not that “results showed 45 they possess good cooling and lubricating properties” revise.

Avoid the use of grouped references, instead, cite each author/reference contribution.

Again, lack of cohesion, jumping from cutting fluid costs to EP.

The cohesion between the ideas must be checked in the whole manuscript, it will be no further specifically mentioned in the revision to make it more concise.

Third paragraph, the data of the other studies presented are incomplete, please, make sure that at least the cutting parameters, tool characteristic and cutting fluid data are available. Also, avoid global terms such as “produced the best overall results”, “made little contribution 61 to surface roughness and tool life”, “tool wear and chip thickness can be 64 reduced by using nano-cutting fluid”. Detail the results more, use numerical data to corroborate to what you are presenting, if the mentioned paper do not present numerical data, add their hypothesis.

Line 71, what is considered a “Reasonable use of cutting fluid”? double check all the manuscript to avoid the vague terms.

Line77 there is also a missing period symbol.

Differentiate LQL to MQL.

First paragraph in page 3, same thing with the data.

Materials and methods

What heat treatment was applied which mechanical properties it generates in the alloy? (at least the hardness should be presented).

Was the tool coated?

Check the grammar use of the word kind “which is kind of water-soluble”.

Check the grammar use of the word ingredients (I believe components or chemical components will be more adequate).

Check the correct tense, usually the methodology is presented in the past, example: “The three-axis machining center (TV600, CNC, China) is used,” to “The three-axis machining center (TV600, CNC, China) was used,” Check it in the complete manuscript.

Table 3: maintain the same number of significant algharisms (check in the whole manuscript).

How were the cutting parameters selected? By the literature? preliminary tests?

Which interval was used? Give the direct value in the methodology. (line 138).

What is the operation range of the dynamometer? Which frequency was used to obtain the data? Is the frequency used in the operation range of the dynamometer ? The tool rotation was about 15000 rpm, probably it is in the range for the equipment used, but it need to be detailed.

Also, was any filter used to treat the obtained force signal? Detail the conditioning parameters.

The equipment needs to be better defined. Which SEM was used? Which is the precision of the TV600? And the spindle speed? Axis resolution? Same thin for all the equipment used.

Results

3.1.1. Reference the viscosity definition

Add a more detailed comment on the result, after line 152 each one would be better for the aluminum milling?

Also, it would be interesting to compare the result with literature results for cutting fluid viscosity. The results must be discussed and not only presented.

Same thing of the significant algharisms on the fig. 2. Check all the figures

3.1.2 Better detail on the results than item 3.1.1, still needing the discussion with literature results. It should be applied for all the presented results of item 3.1 and 3.2. it is not adequate for a research to not compare the results with the literature.

3.2.2

Could the data be presented also in terms of surface roughness? It would be better for comparison, not only for being an numerical data but for comparison if the surface generated in this work can be considered good.

Usually the scanning software make it available.

3.3.1

The cutting force can influence in the surface and not only on tool life.

It is important to define how the forces were measured for the reader to know which one is the cutting force, which is feed…

The force signal seems to present noise. If the filters and data of the dynamometer were mentioned one would be able to know if it is pure noise or really a machining characteristic.

In fig. 8 a, where is the fx in the graph?

Put similar graphs in the same scale.

How did the author correlate the force results with vibration? Was any analysis in the frequency?

All the force results needs to be revised, better defined and compared to the literature.

If the wear was measured in only 6 different intervals how was the wear curve obtained?

How did the first tooth not worn out for QC-2803?

Chipping is not the only tool wear mechanism that can be observed in fig. 13, it needs to be better discussed.

There is no clear third wear stage for any of the cutting conditions.

Discussion

-          Use more values to discuss, not only, more, better, higher…

-          Revise the discussion after doing the suggestions about the results. Add more literature to compare.

Conclusions

-          Not proper to say “his will dramatically reduce the vibration between 316 the tool and the workpiece.” It is not dramatic and you did not measure vibration or indicated how you obtained it indirectly by the dynamometer.

-          Mention the specific gains on the physical and chemical properties of the cutting fluid.

-          How does there is a conclusion on the surface roughness if it is not presented as a result???????????

-          Revise it all after the review of the other parts.

The manuscript is about the application of a new cutting fluid to mill aluminum alloys, each is relevant to the field. However, it needs some improvements to be adequate for publishing.

Reviewer 3 Report

My opinion is major, but encourage authors to provide a better version.

Reviewer 4 Report

Manuscript Number: Materials-2236398

Title: Effect of Cutting Fluid on Milled Surface Quality and Tool Life of Aluminum Alloy

Decision: Minor revision

Article Type: Article

The article is, in general, well written but there are some issues that article should consider to revise in order to improve its quality. Some comments were done in this way:

·       Fig. In the graph in Figure 6, the surface roughness of CCF-10 decreases seriously after 150 min. It is a known fact that increasing tool wear over time has significant effects on surface roughness. How do the authors interpret the decline here?

·       Why are the wear values given as 0 in the values given in Table 5? The actual wear values here should be given even if they are low.

·       Figure 11 SEM images should be taken even closer and placed next to them as in Figure 12. Because the wear limits and types are not understood. In addition, the abrasions should be detailed by drawing.

After making the above corrections would recommend this article for publication in Materials

Round 2

Reviewer 1 Report

MANUSCRIPT ID - materials-2236398

The authors have not responded to the previous queries that I had raised. However, based on the changes made in the manuscript with respect to the other reviewers’ comments, I have suitably modified my earlier comments and the same are provided below: 

1. The authors have a number of new references. However, many of them in the introduction section appear irrelevant to the main aim of this paper which is to compare two different cutting fluids. The authors should cite works that have compared different cutting fluids, their properties and their effect on machinability. There is no need to present works relevant to MQL, LQL, nanofluids, compressed nitrogen gas, cooling lubrication techniques, etc. which are separate areas of research. References similar to [33], [34], which deal with cutting fluids of different composition should be added.

2. “The tool used in the study was an uncoated three-edge straight shank carbide end milling cutters (XS2111-075K-GNS, SG, China) with a diameter of 10 mm and a radius of 0.25mm.” - Please check the radius value.

3. “A novel external circulation system was designed for the convenient replacement of cutting fluids.” Choice of better words is needed. “Replacement of cutting fluids” does not sound appropriate.

4. What is the reason for reduction in surface roughness with machining time? The authors can compare the surface morphology at the beginning and end of machining machining time for the two fluids.

5. What is the reason for using the words “real cutting force”? If it is the resultant cutting force, how can the amplitude of the resultant force be 80 N when the maximum force in the X-direction and Y-direction be 80 N and 50 N respectively for QC 2803? 

Reviewer 3 Report

My comments were taken into account. Accepted.

Author Response

Many thanks.